Journal of Data-centric Machine Learning Research (2025)          Submitted 6/25; Published 9/25

# MM-Gen: Principled and Generalizable Data Curation for Enhancing Task Performance in VLMs

**Siddharth Joshi**                                    SJOSHI804@CS.UCLA.EDU
*Microsoft Research and*
*University of California, Los Angeles*

**Besmira Nushi**                                      BNUSHI@MICROSOFT.COM
*Microsoft Research*

**Vidhisha Balachandran**                              VIDHISHAB@MICROSOFT.COM
*Microsoft Research*

**Varun Chandrasekaran**                               VARUNC@MICROSOFT.COM
*Microsoft Research and*
*University of Illinois Urbana-Champaign*

**Vibhav Vineet**                                      VIBHAV.VINEET@MICROSOFT.COM
*Microsoft Research*

**Neel Joshi**                                         NEEL@MICROSOFT.COM
*Microsoft Research*

**Baharan Mirzasoleiman**                              BAHARAN@CS.UCLA.EDU
*University of California, Los Angeles*

**Reviewed on OpenReview:** *https://openreview.net/forum?id=r9xVVuhxKI*

**Editor:** Sergio Escalera

**Keywords:**   Vision-Language Models, Data Curation, Task Performance, Synthetic Data Generation

## Abstract

Vision-language models (VLMs) often struggle on specialized tasks requiring fine-grained image understanding due to inadequate task-specific text annotations in the training data. We introduce MM-GEN, a framework for data curation that improves VLM performance on such tasks guided by four principles: coverage of task subgroups, diversity of examples, quality of annotations, and informational value. Given reference samples from the target task, keywords enumerating task subgroups, and a pool of candidate images, MM-GEN implements a multi-stage process: (1) partitioning data by subgroup to ensure coverage, (2) generating diverse annotations via in-context learning for each subgroup using corresponding reference samples, and (3) applying perplexity-based filtering to ensure high quality annotations while prioritizing examples that provide novel information to the model. When fine-tuning Llava-1.5 (7B) with our generated data, we achieve absolute improvements of 15%, 14%, and 29% on chart understanding, diagram interpretation, and spatial reasoning tasks, respectively. Moreover, our filtering approach enables discarding 50% of the data without performance loss. Our results confirm that task-specific text curation is indeed the critical bottleneck in VLM performance, and MM-GEN provides a principled and generalizable solution that

can be applied to any image-understanding task with minimal human intervention. Code available at `https://github.com/sjoshi804/MM-Gen`.

## 1 Introduction

Although vision language models (VLMs) excel at many multimodal tasks (Liu et al., 2023), they often struggle with more complex challenges requiring fine-grained understanding of image details (Balachandran et al., 2024; Fu et al., 2024; Kamath et al., 2023). We argue this limitation stems from training data quality; while VLMs are trained on rich web-scraped images, the accompanying text descriptions frequently lack relevance to the image (Nguyen et al., 2024) or omit crucial details necessary for complex reasoning (Lai et al., 2024a). Fig. 1 demonstrates this problem with examples where web captions fail to capture essential information for chart understanding, spatial reasoning, and diagram interpretation tasks.

Recent work has addressed data quality through synthetic caption generation (Nguyen et al., 2024; Lai et al., 2024a; Yu et al., 2024). However, these approaches remain task-agnostic and cannot guarantee that task-relevant details are preserved. Shi et al. (2024) manually curated a dataset for multimodal mathematical question-answering by enhancing existing data with detailed annotations using strong VLMs, but such manual curation—including sourcing questions from high-quality mathematical multimodal datasets and crafting specific prompts to diversify them—requires substantial human effort and domain expertise, limiting scalability across diverse applications (Masry et al., 2024; Zhang et al., 2024).

To address these limitations, we introduce MM-Gen, a principled framework for automatically generating task-relevant text annotations for images with minimal human effort. By automating this process in a generalizable way, MM-Gen enables VLMs to perform better across specialized tasks—an essential step toward their broader deployment and adoption. Our approach is guided by key principles established in prior data curation literature (Muennighoff et al., 2025; Joshi and Mirzasoleiman, 2023; Mirzasoleiman et al., 2020): (1) **Coverage**—ensuring all task-relevant subgroups are represented in the training data; (2) **Diversity**—incorporating varied examples to represent each subgroup; (3) **Quality**—ensuring examples contain accurate and coherent information; and (4) **Informativeness**—prioritizing examples that provide novel information to the model. Achieving each criterion presents significant challenges. As established in (Rolf et al., 2021; Shahbazi et al., 2023), ensuring comprehensive subgroup representation remains particularly difficult. Moreover, achieving genuine diversity with synthetic data generation has been a persistent challenge across modalities and training algorithms (Chang et al., 2023; Zhu et al., 2025; Norman and Whitney, 2024; Rotstein et al., 2023; Lai et al., 2024b; Fan et al., 2024; Yu et al., 2024). Finally, ensuring quality and informativeness of datasets in open problem in machine learning (Mindermann et al., 2022; Joshi and Mirzasoleiman, 2023; Zhao et al., 2024) and hasn't been tackled before for VLMs.

MM-Gen achieves these 4 desiderata through a multi-stage process requiring: (i) a small set of reference samples from the target task, (ii) a list of associated image types (subgroups), and (iii) a pool of candidate images. The framework first partitions both reference examples and candidate images by subgroup, ensuring **coverage** across all categories. For each subgroup, MM-Gen generates tailored text annotations by conditioning a strong VLM on randomly sampled reference examples, yielding greater **diversity** than traditional natural language

prompting. MM-Gen then applies perplexity-based filtering to remove both high-perplexity outliers (likely incoherent or incorrect) and low-perplexity examples (providing minimal learning signal), thus ensuring **quality** while maximizing **informativeness**. This systematic and generalizable approach enables significant performance improvements on *any image understanding task.*

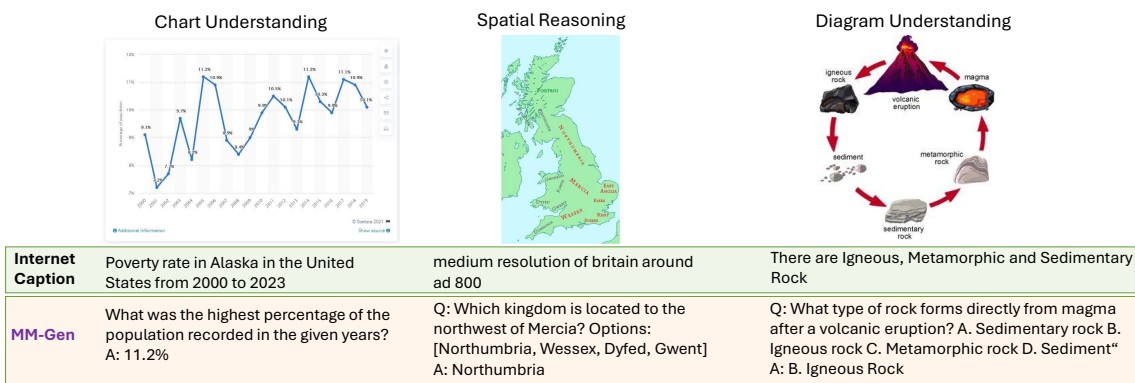

Figure 1: Examples of general text captions vs. task-specific text annotations generated by MM-Gen and used for fine-tuning supervision.

We evaluate MM-Gen on three challenging fine-grained image understanding tasks: chart understanding, diagram interpretation, and spatial reasoning on maps. Using Llava-1.5 as our base model, MM-Gen enables absolute improvements of 15%, 14%, and 29% on these tasks for the 7B parameter version. The gains extend to larger models (Llava-1.5 13B) as well, as we see absolute improvements of up to 20% across tasks. Moreover, our perplexity-based filtering reduces data volume by up to 50% while maintaining or improving performance, demonstrating its effectiveness for training VLMs. Models trained with MM-Gen data consistently outperform those using generic captions or annotations generated without task-specific references. We also conduct ablation studies on reference sample set size, subgroup partitioning, and in-context sample scaling.

In summary, our contributions are:

1. Demonstrating task-aware text annotations significantly outperform task-agnostic approaches for VLM finetuning

2. Establishing that specifying the task in a data-centric manner (i.e. using reference examples), rather than natural language instructions, better achieve coverage and diversity in generated annotations

3. Showing perplexity-based filtering effectively balances data quality and efficiency

4. Creating a scalable pipeline that improves VLM performance by up to 30% across multiple tasks with minimal human intervention

## 2 Related Work

**Synthetic Data Generation for Multimodal Models:** Existing approaches to synthetic data generation for VLMs fail to explicitly target all four critical desiderata: *coverage*, *diversity*, *quality*, and *informativeness*. Nguyen et al. (2024) highlighted the low quality of web-scraped captions, demonstrating their inadequacy for tasks requiring fine-grained visual understanding. Subsequent approaches such as synthetic caption generation (Rotstein et al., 2023; Lai et al., 2024b; Fan et al., 2024; Yu et al., 2024) typically focus on prominent objects but lack the necessary *coverage* of task-specific details. Recent work combining real and synthetic data using stronger VLMs (Li et al., 2023a; Chen et al., 2023b; Liu et al., 2023) achieve limited *diversity* but depend heavily on human expertise for prompt engineering, hampering scalability. MiniGPT-4 (Zhu et al., 2023) attempts to improve *quality* through strong VLMs but relies on labor-intensive manual filtering that becomes impractical at scale. Task-specific approaches like MathLLava (Shi et al., 2024) and ChartInstruct (Masry et al., 2024) require substantial human oversight to ensure adequate *quality* and *coverage*, but their specialized nature restricts generalizability. Critically, no existing method systematically addresses *informativeness* by identifying the examples that provide the strongest learning signal. In contrast, MM-GEN addresses all four critical criteria: (1) comprehensive *coverage* of task-relevant details, (2) sufficient *diversity* in generated annotations, (3) effective *quality* control without manual intervention, and (4) *informativeness* through principled filtering.

**Synthetic Data Generation for Training LMs:** Recent works (Eldan and Li, 2023; Gunasekar et al., 2023; Li et al., 2023b; Abdin et al., 2024; Mukherjee et al., 2023; Dubey et al., 2024) demonstrated LMs' effective pre-training on synthetic data, while (Mitra et al., 2023, 2024) highlighted synthetic task-specific data's efficacy for specialized tasks. These approaches focus exclusively on text, neglecting multimodal data generation's unique challenges, particularly ensuring comprehensive *coverage* of task-relevant visual details through the text annotations. MM-GEN is the first such method for VLMs.

**Data Filtering Methods** Filtering techniques span supervised learning (Coleman et al., 2019; Toneva et al., 2018; Swayamdipta et al., 2020; Paul et al., 2021; Katharopoulos and Fleuret, 2018; Mirzasoleiman et al., 2020; Pooladzandi et al., 2022; Killamsetty et al., 2021), self-supervised learning (Joshi and Mirzasoleiman, 2023; Tripathi et al.), multimodal contrastive learning (Joshi et al., 2024; Evans et al., 2024; Fang et al., 2023; Abbas et al., 2023; Maini et al., 2024), and generative LMs (Marion et al., 2023; Tirumala et al., 2023; Zhou et al., 2024; Chen et al., 2023a; Yang et al., 2023b). However, these have not been applied to filtering multimodal generative data for VLMs. We adapt filtering from (Marion et al., 2023) to discard up to 50% of generated data while maintaining performance.

## 3 Problem Formulation

Our objective is to generate text annotations for a given pool of candidate images, to improve performance, of a given VLM, on a target task $T$. Let a multimodal sample be denoted as $s = (v, t)$, where $v$ represents an image and $t$ represents the associated text (both the text prompt and text response). Let $V_T^{pool}$ denote the provided pool of candidate images, e.g., a corpus of chart images, and $N_{gen}$ the number of multimodal samples we wish to curate. Let $S_T^{ref}$ be a small ($|S_T^{ref}| = n \ll N_{gen}$) set of *reference samples* that is *representative* of the task

$T$. This set serves as a reference for the text that is relevant for task $T$. In practice, this could be samples from the validation set of a dataset for chart understanding like ChartQA (Masry et al., 2022). Additionally, let types$_T$ denote a list of the types of images associated with the task. For tasks like chart understanding, which have several different types of images, types$_T$ could include *bar charts*, *line charts*, and *pie charts*. The goal then is to use $S_T^{\text{ref}}$, $V_T^{\text{pool}}$, and types$_T$ to generate $N_{\text{gen}}$ multimodal samples for fine-tuning a given VLM, to improve performance on task $T$. To generate annotations, we assume access to a stronger VLM, i.e., one with higher performance than the given VLM on target task $T$ [1].

## 4 MM-Gen Overview

In this section, we first motivate the need for task-specific text annotations through an empirical case study. We then present MM-Gen: our framework for generating task-specific text annotations that satisfy all four desiderata for effective dataset curation: coverage, diversity, quality, and informativeness. Importantly, MM-Gen is designed to generalize to any task with minimal human supervision.

### 4.1 Challenge of Coverage in Task-Agnostic Text Annotations (Case Study on MS COCO)

We demonstrate how even high-quality human-crafted text annotations can fail to provide adequate *coverage* of visual elements critical for downstream tasks. MS COCO (Vinyals et al., 2016) is widely regarded as a high-quality, large-scale dataset commonly used for training image captioning models (Santurkar et al., 2023; Nguyen et al., 2024). Each image includes 5 manually crafted descriptive captions, with annotators explicitly instructed to describe 'all relevant details.' Despite these rigorous annotation guidelines, Figure 1 reveals *significant*

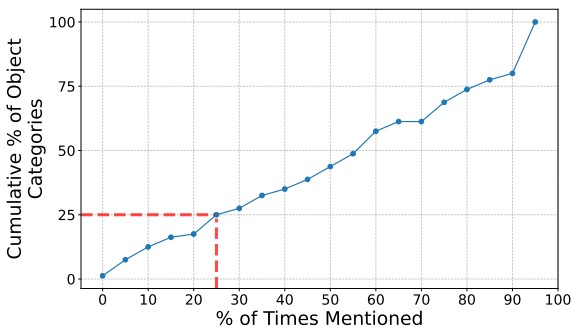

Figure 2: Even high-quality human-curated captions (MS COCO) have poor coverage for many visual details

*gaps in coverage.* Notably, 25% of object categories appearing in images are mentioned in the accompanying captions only 25% of the time (meaning they are omitted 75% of the time). For downstream tasks that depend on recognition of these under-represented

---

1. In practice, this can be a VLM specialized on the task of interest (e.g., a VLM specialized for object detection if the task is detection), a general stronger model than the model of interest or a combination of these.

categories, even MS COCO's "high-quality" captions provide inadequate supervision. This analysis highlights that carefully curated but task-agnostic text annotations frequently miss information important for tasks requiring specific visual details. This is not simply a long-tail problem (Changpinyo et al., 2021); these visual elements are present in many images but omitted in the corresponding text. For example, descriptive captions of charts might thoroughly describe their general appearance but omit crucial details like minimum/maximum values, temporal trends, or specific data points—precisely the information needed for chart understanding tasks.

## 4.2 MM-Gen: Design

We now introduce MM-GEN: an automated framework to generate text annotations that provide task-specific supervision while satisfying all four key dataset curation principles. Using chart understanding as exemplified by the ChartQA (Masry et al., 2022) dataset as our running example, we describe how MM-GEN addresses each of the four desiderata. Our goal is to improve Llava-1.5-7B (Liu et al., 2023) using a stronger VLM such as GPT-4 (OpenAI, 2023). The inputs to MM-GEN are:

1. Reference Sample Set $S_T^{ref}$: Examples from the ChartQA validation set

2. Types of Images $types_T$: [`'bar chart'`, `'pie chart'`, `'line chart'`]

3. Candidate Image Pool $V_T^{pool}$: Corpus of chart images containing bar charts, pie charts, and line charts

### 4.2.1 COVERAGE: PARTITIONING DATA INTO SUBGROUPS

**Problem** Multimodal tasks often span diverse image types, each requiring attention to different visual elements. Generic approaches to dataset curation frequently fail to adequately cover all task-relevant visual details across the full spectrum of image types. Generating text annotations without considering image subgroups can lead to uneven representation and neglect of critical visual elements in certain image categories.

**Solution: Partitioning Data into Subgroups** To address this coverage challenge, we partition both the reference sample set and candidate image pool into distinct subgroups based on the image types specified as $types_T$ before generating text annotations. This ensures that MM-GEN explicitly generates text annotations for all subgroups in the downstream task, guaranteeing coverage. We rely on predefined keywords in $types_T$ rather than automated image clustering for this partitioning. This design choice is deliberate: direct image clustering can lead to groupings based on spurious correlations (Yang et al., 2023a) (e.g., a bar chart and a pie chart with similar color schemes might be grouped together). Such correlations, while visually apparent, are not semantically meaningful for the task and will not guarantee coverage of task-relevant subgroups. By using explicit type keywords, we ensure the partitioning aligns with semantically meaningful distinctions that matter for the downstream task.

**Implementation** We leverage CLIP's (Radford et al., 2021) zero-shot classification capabilities to partition images according to $types_T$. We encode texts from $types_T$ with CLIP's text encoder $f_T$ and images from $S_T^{ref}$ and $V_T^{pool}$ with its vision encoder $f_V$. Each image is

assigned to the text category with the highest cosine similarity:

$$k^* = \arg \max_{k \in \text{types}_\text{T}} S_C\big(f_V(v), f_T(k)\big)$$

where $v$ represents the image and $k$ refers to the $k$-th text in $\text{types}_\text{T}$. This produces partitioned reference samples and candidate pools:

$$\text{S}_\text{T}^\text{ref} = \bigcup_{k \in \text{types}_\text{T}} \text{S}_\text{T}^\text{ref}{}_k, \quad \text{V}_\text{T}^\text{pool} = \bigcup_{k \in \text{types}_\text{T}} \text{V}_\text{T}^\text{pool}{}_k$$

### 4.2.2 Diversity: Generating Text Annotations Using Reference Samples

**Problem** Fixed natural language instruction templates inevitably lead to homogeneous text annotations that fail to capture the full range of ways task-relevant information might be expressed. This lack of diversity limits the robustness of models trained on such data, as they may overfit to a limited set of annotation patterns and fail to generalize.

**Solution: Reference-Based Text Annotation Generation** Rather than relying on natural language task descriptions, we employ a data-centric approach that leverages the diversity of a representative set of reference samples to specify the task to a stronger VLM. By prompting the VLM with varied reference samples, we ensure the generated text annotations are diverse. Importantly, this significantly reduces human effort—rather than crafting highly detailed natural language instructions, simply selecting a small number of reference samples (e.g., a subset of the validation set) is sufficient.

**Implementation** For each subgroup $(\text{S}_{\text{T}_k}^\text{ref}, \text{V}_{\text{T}_k}^\text{pool})$, we generate text annotations by randomly sampling a reference sample from the subgroup to serve as an in-context learning example alongside a candidate image. For each subgroup, we generate a fraction of the target dataset size $\text{N}_\text{gen}$ proportional to that subgroup's representation in the reference set, thereby maintaining the natural distribution of the task while ensuring diverse examples within each category.

### 4.2.3 Quality and Informativeness: Filtering using Perplexity

**Problem** A key challenge in using a stronger VLM to generate training annotations lies in ensuring both the *quality* and *informativeness* of the resulting dataset. Despite being more capable, the stronger VLM may still produce malformed, incorrect, or low-quality examples, which can degrade the training signal and ultimately harm downstream performance. Simultaneously, many generated annotations may already be correctly handled by the VLM we aim to improve, making them uninformative for training and leading to inefficient use of computational resources.

**Solution: Perplexity-Based Filtering** Prior work has typically relied on manual filtering or highly specialized prompts for LLM-Judges (Shi et al., 2024) to ensure the quality of generated datasets, with less focus on maximizing informativeness. However, manual curation requires significant human effort and becomes prohibitively expensive at scale, while designing effective prompts for LLM-judges demands considerable expertise and still requires multiple resource-intensive model runs. Instead, we address both quality and informativeness simultaneously using lightweight *perplexity-based filtering*, drawing inspiration from techniques used in LLM pre-training (Marion et al., 2023). Perplexity

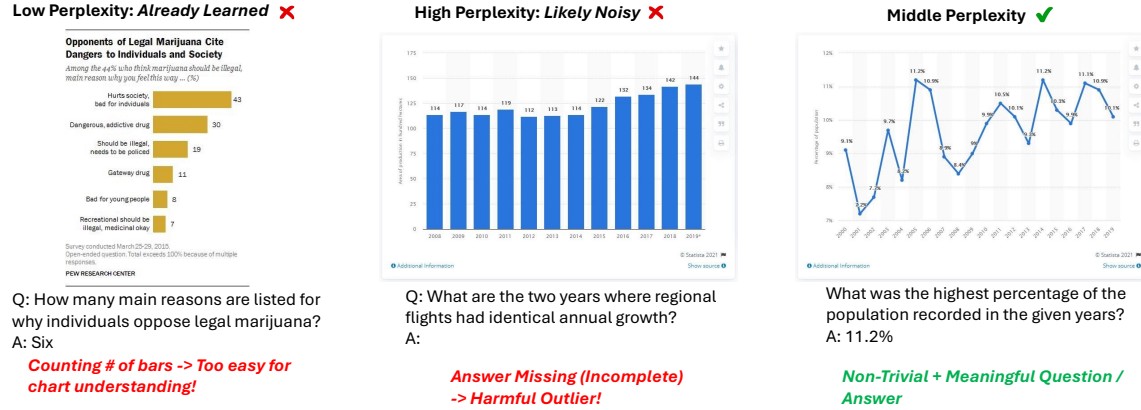

Figure 3: Examples of different text perplexity values mapped to easy cases (low perplexity), outliers that are likely malformed or incorrect (high perplexity), and meaningful, non-trivial questions (middle perplexity). Questions with middle perplexity are most likely to provide 1) informative and 2) high-quality training signal.

measures how well an auto-regressive model predicts a given sequence of tokens. Intuitively, it captures the model's uncertainty or surprise over the generated text. Our approach retains only examples with middle perplexity as measured by the VLM we are training. Examples with high perplexity are, by definition, those that the model finds highly improbable—often due to being outliers or containing malformed or incorrect content. Since the stronger VLM is generally reliable, such high-perplexity cases are likely artifacts or noise in the data. Discarding these examples helps ensure the resulting dataset is of higher quality. Conversely, examples with low perplexity correspond to prompts and answers that the current (weaker) VLM can already predict confidently. These examples provide little new learning signal and are thus uninformative for training. Removing them improves efficiency by focusing training on more challenging, beneficial cases.

As illustrated in Figure 3, *middle-perplexity examples* strike the right balance: they are difficult enough to challenge the current model and provide new learning signal, yet not so difficult that they are likely to be erroneous. These examples are thus the most valuable for fine-tuning.

**Implementation** Perplexity is defined as

$$\exp\left(-\frac{1}{n}\sum_{i=1}^{n}\log P(w_i \mid w_1, \ldots, w_{i-1})\right).$$

For each generated example, we compute the perplexity of the text response, conditioned on the image and text prompt, using the VLM we wish to improve. Empirically, we retain 50% of the generated data (the middle-perplexity examples), which demonstrates significant gains in both performance and efficiency across tasks.

Through this systematic, principled approach to dataset curation that directly addresses each desideratum, MM-GEN satisfies all four key requirements—*coverage*, *diversity*, *quality*, and *informativeness*—resulting in synthetic datasets that enable significant performance improvements across diverse visual understanding tasks.

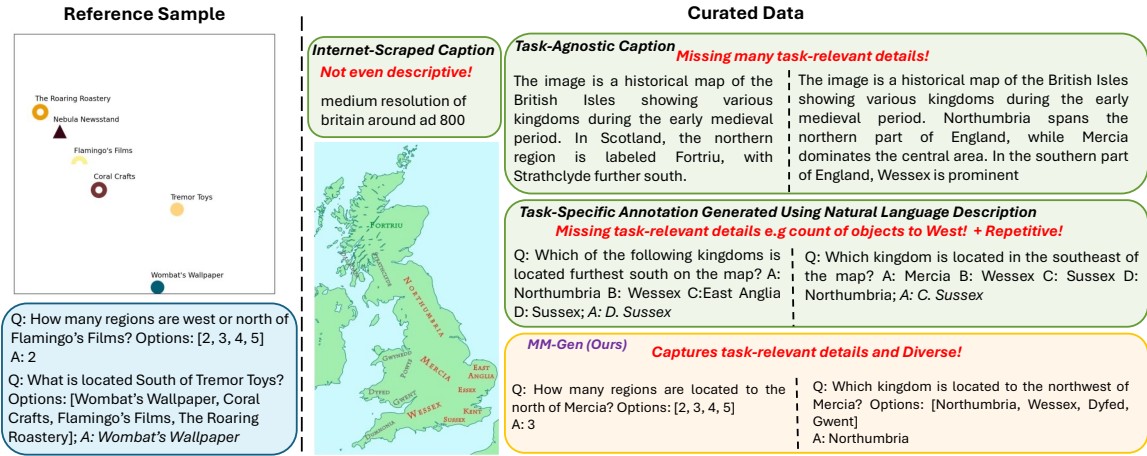

Figure 4: Comparing different baselines for multimodal data generation with MM-Gen. MM-Gen not only customizes the generated text to the task via reference samples, but it also adds missing details to the text that are required for answering the task.

## 5 Experiments

**Tasks.** We evaluate MM-Gen on 3 complex multimodal tasks, requiring fine-grained understanding of details in the images, that several existing VLMs struggle on: 1) chart understanding & reasoning, 2) diagram understanding, and 3) spatial reasoning on maps. Although we evaluate MM-Gen on these three currently challenging tasks, it is a general method that can be applied to any visual understanding task without modification.

*Chart Understanding and Reasoning:* We use ChartQA (Masry et al., 2022) to evaluate the ability of a model to understand and reason over chart-based visualizations. As inputs to MM-Gen, we have: 1) Reference Samples: the validation set of ChartQA ($\approx$ 1K samples); 2) Types of Image: determined from dataset description as [`bar chart`, `line chart`, `pie chart`]; 3) Candidate Image Pool: 15K images of charts taken from the ChartQA training set. With these inputs, we curate 150K multimodal samples and retain 75K after filtering.

*Diagram Understanding:* We use AI2D Diagrams (AI2D) (Kembhavi et al., 2016) to asses a model's diagrammatic understanding using grade-school science diagrams and associated multiple-choice questions about the relationships and components in these diagrams. As inputs to MM-Gen, we have: 1) Reference Samples: a random subset of size 100 sampled from AI2D's training set; 2) Types of Image: determined as [`physics diagram`, `biology diagram`, `chemistry diagram`, `geography diagram`] from the dataset description; 3) Candidate Image Pool: approximately 5K diagram images taken from the training images of AI2D. With these inputs, we curate a total of 100K multimodal samples and retain 50K after filtering.

*Spatial Reasoning on Maps:* We use SpatialMap (Wang et al., 2024) to test the spatial reasoning capabilities of VLMs on maps by requiring them to answer questions on cardinal directions (e.g., North, South, East, West) and reasoning about the relationships between different landmarks in the map. As inputs to MM-Gen, we have: 1) Reference Samples:

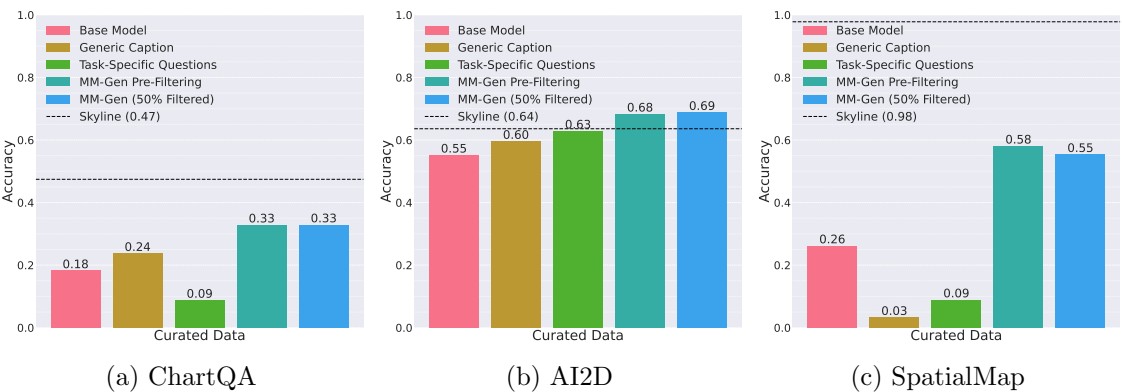

Figure 5: Comparison of MM-Gen performance across tasks against contributed baselines and skyline.

the validation set of SpatialMap; 2) Types of Image: determined from dataset description as ['map']; 3) Candidate Image Pool: 1K images of maps retrieved from DataComp-Small (Gadre et al., 2024) using CLIP embedding search. With these inputs, we curate 50K multimodal samples and retain 25K after filtering.

## 5.1 Analysis of Performance of across Tasks

**Baselines and Skyline.** Since MM-Gen is the first framework for curating task-specific multimodal samples, we contribute baselines and a skyline to evaluate its effectiveness. We use GPT-4o (OpenAI, 2023) as the stronger VLM to generate the text annotations. Exact inputs and generated examples appear in App. A. We enumerate them below:

*1. Base Model*: This refers to the initial performance of the VLM, before any additional training.

*2. Task-Agnostic Captions*: This baseline uses task-agnostic text annotations generated by a stronger VLM for the candidate image pool. This tests the importance of *coverage* of task-relevant details, as traditional caption generation methods do not specifically target the details needed for the target visual understanding tasks.

*3. Task-Specific Text Annotations (No Reference Images)*: This baseline uses text annotations generated by a stronger VLM to be task-specific using a natural language description of each task. These descriptions are obtained from the original dataset descriptions (?Kembhavi et al., 2016; Wang et al., 2024). This comparison tests the effectiveness of achieving *diversity* and *coverage* through natural language instructions versus reference samples (MM-Gen).

*4. Skyline – Training on i.i.d. Training Data*: The skyline refers to i.i.d. training data (curated manually by humans) specifically for the target task that includes task-relevant details; it provides a performance benchmark for MM-Gen to approach or surpass by representing the upper bound of what can be achieved with high *coverage*, *diversity*, *quality*, and *informativeness* in the training data. Each of the three tasks we consider, in addition to test data, also provides i.i.d. training data, which we use as the skyline. For (1) ChartQA, this consists of ∼30K chart images with associated chart understanding question-answers; for (2) AI2D, ∼ 5K grade school diagram images paired with diagram understanding questions;

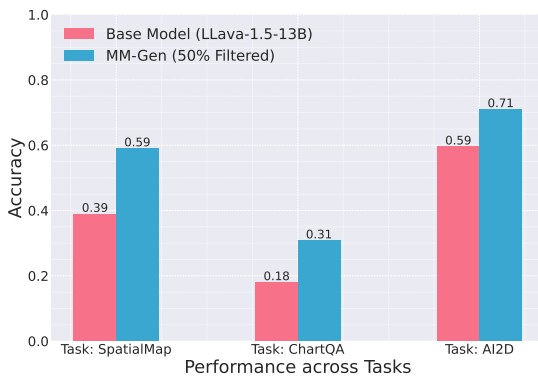

Figure 6: Evaluation on Llava-1.5 (13B Parameters)

and for (3) SPATIALMAP, we generate ∼15K map images paired with spatial reasoning questions-answers using the test provided data generation scripts.

**Models.** As the target VLM to improve, we use Llava-1.5 (7B parameters) (Liu et al., 2023), comparing the performance of the base model (before training on any additional data) to that of training on the data curated by the aforementioned baselines, the skyline and MM-GEN. To investigate the effectiveness of our approach across model sizes, we additionally evaluate MM-GEN on Llava-1.5 (13B parameters). Further details in Appendix D.

Fig. 5 shows that MM-GEN significantly improves upon the base model across all three tasks, and either closes the gap with or surpasses the skyline performance. On CHARTQA, MM-GEN achieves an absolute improvement of 15% over the base model, reaching 0.5× of the skyline's gain. On AI2D, it achieves a 14% absolute improvement and exceeds the skyline, achieving 1.6× the improvement that the skyline provides. Finally, on SPATIALMAP, MM-GEN shows a 29% absolute improvement over the base model, reaching 0.4× of the skyline's gain. A qualitative comparison of all baselines is shown in Fig. 4.

Across all three tasks, MM-GEN consistently outperforms baseline 2 (task-agnostic captions), underscoring the critical importance of *coverage*—i.e., including task-relevant details in the annotations. Qualitative examples in Fig. 4 reveal that task-agnostic captions often overlook features essential for specialized tasks such as chart understanding or spatial reasoning. MM-GEN also outperforms baseline 3 (task-specific annotations derived from natural language instructions), further emphasizing the value of specifying the task in a data-centric manner—namely, through reference samples—rather than relying solely on textual descriptions. The shortcomings in baseline 3 highlight how difficult it is to capture the full scope of task-relevant information via natural language alone.

In addition to coverage, we also observe substantial improvements in *diversity*. Manual inspection of the generated data reveals that both baselines suffer from limited diversity in their annotations (cf. Appendix C). Despite being task-aware, baseline 3's instructions yield annotations that are often repetitive and formulaic, failing to capture the full range of variation and nuance inherent in the tasks. This lack of diversity can lead to overfitting, ultimately reducing the model's ability to generalize. In contrast, MM-GEN generates

diverse annotations that reflect the variability present in the reference sample set, resulting in better generalization across tasks.

Thus, MM-Gen is not only more effective but also more adaptable, as it does not require extensive human effort to craft detailed task descriptions. Notably, on both ChartQA and SpatialMap, baselines 2 and 3 even degrade performance relative to the base model—an outcome we attribute to their poor **coverage** and **diversity**.

Beyond coverage and diversity, MM-Gen also excels in ensuring high **quality** and **informativeness** through its perplexity-based filtering strategy. Across all three tasks, the 50% filtered MM-Gen dataset achieves performance nearly equivalent to that of the full, unfiltered dataset—while requiring only half the training resources. This demonstrates that our filtering strategy effectively removes predominantly uninformative or low-quality examples. Notably, on AI2D, we observe a slight performance gain after filtering, highlighting the ability of MM-Gen's filtering to identify higher-quality data that can outperform even a 2x larger, less curated set. On SpatialMap, the modest 3% drop in performance in filtered data performance can be attributed to the higher diversity in MM-Gen's unfiltered dataset. This diversity arises from the nature of the task, where questions involving pairs or groups of objects scale combinatorially with the number of objects on the map, allowing for significant diversity in MM-Gen's unfiltered generated text annotations.

Finally, we note that the magnitude of MM-Gen's absolute improvements over the base model varies across tasks, which can be attributed to the differing levels of difficulty the base model faces in each setting. This is reflected in the wide range of base accuracies observed. The relative improvements compared to the skyline also vary, primarily due to differences in the size and quality of the skyline datasets. For example, on SpatialMap, the skyline performance is near-perfect, as the skyline data is created programmatically using the same code used to generate the test set and is thus perfectly i.i.d. In contrast, on AI2D and ChartQA, where data is curated by humans, the correspondence between training and test data is necessarily weaker. Moreover, the AI2D skyline dataset is relatively small ($\approx 5K$), which may contribute to its limited improvement. Despite these differences, MM-Gen consistently narrows the gap to skyline performance, demonstrating that for real-world tasks, it can curate task-relevant training data that is nearly as effective as human-curated datasets—while requiring only minimal human effort i.e. collecting a small set of reference samples, candidate images, and determining the image types.

Fig. 6 shows that, across all tasks, MM-Gen can even improve models as large as Llava-1.5 (13B Parameters). In fact, the resulting performance, across tasks, is even higher than that achieved by Llava-1.5 (7B parameters) in Fig. 5. This shows that MM-Gen curated data can help boost performance of relatively stronger VLMs as well, utilizing their superior initial performance to achieve even higher performance, on target tasks.

**Performance on Control Tasks** In Table 1, we show that training on MM-Gen data, to improve performance on a given target task, does not affect performance on other tasks (control tasks). Here, we use MMMU (Yue et al., 2024) to represent these tasks as it is considered a comprehensive evaluation of VLMs across many domains.

**Combining Data from All Tasks** We also consider training Llava-1.5 (7B) in Table 2 on a combination of data generated by MM-Gen for all tasks and observe that it can simultaneously improve performance across all three tasks. This demonstrates how MM-Gen can be

Table 1: Effect of Performance on Control Tasks (MMMU)

| Model | Accuracy (%) |
|---|---|
| Base Model | 35.8 |
| MM-Gen (ChartQA) | 33.6 |
| MM-Gen (AI2D) | 37.0 |
| MM-Gen (SpatialMap) | 34.1 |

Table 2: Performance of Training on Combined MM-Gen Data.

| Model | Base Model (%) | MM-Gen All (%) |
|---|---|---|
| ChartQA | 18.2 | 25.9 |
| AI2D | 55.2 | 65.7 |
| SpatialMap | 18.2 | 44.2 |

used to design datasets that achieve better coverage, diversity, quality and informativeness, to train holistically more performant VLMs.

## 5.2 Ablations

Here, we conduct ablations for MM-Gen on the chart understanding task (ChartQA). We vary different components of text annotation generation, and compare performance training on the resulting data. We do not filter the data here to isolate the differences in text generation.

**Importance of Partitioning into Subgroups**: Here, we investigate the importance of the partitioning into subgroups performed by MM-Gen prior to data generation by comparing performance with and without partitioning on ChartQA. As shown in Table 3, partitioning contributes a non-trivial 2% of the total 15% improvement that MM-Gen achieves. This highlights the value of explicit subgroup partitioning in ensuring **coverage** across all task-relevant variations.

**Effect of Number of In-Context Samples**: We assess the impact of varying the number of in-context samples provided to the stronger VLM during generation. As seen in Table 3, increasing the number of in-context samples from 1 to 3 actually decreases the final performance, likely due to the limitations of current VLMs on mutli-image understanding (Meng et al., 2024). This suggests that, currently, for maximizing and **diversity**, a single well-chosen reference example is sufficient and potentially optimal.

**Effect of Reference Sample Set Size**: Here, we compare the performance of MM-Gen using a 10× smaller reference sample set. Table 3 shows that MM-Gen can still achieve

Table 3: Ablation Study on MM-Gen using ChartQA

| Ablation | Accuracy (%) |
|---|---|
| MM-Gen | 33.0 |
| MM-Gen without Partition | 31.6 |
| MM-Gen with 3 In-Context Samples | 30.5 |
| 10× Smaller Reference Set | 32.8 |

nearly identical performance, highlighting how even a very small number of reference data is sufficient to ensure adequate **coverage** and **diversity**. This further demonstrates the efficiency of our approach in minimizing human effort while maintaining effectiveness.

## 6 Conclusion

We introduced MM-Gen, a scalable and fully automated framework for curating task-specific multimodal data to improve small vision-language models (VLMs) on specialized tasks. MM-Gen addresses four key desiderata—**coverage**, **diversity**, **quality**, and **informativeness**—through a multi-stage process that takes in a small set of reference samples, image subgroup labels, and candidate images. It partitions data by subgroup, uses a strong teacher VLM to generate diverse, task-aligned annotations, and applies perplexity-based filtering to retain high-quality, informative examples. MM-Gen delivers up to 29% absolute improvements over the base model and can even outperform human-curated data by 1.6×, highlighting its effectiveness when manual curation is infeasible. These findings emphasize the promise of targeted, automated text enrichment for multimodal learning. Future directions include curriculum-based multi-task training and improving teacher signal via model ensembles and answer verification.

## Broader Impact Statement

This work advances the field of Machine Learning by exploring synthetic data generation techniques to enhance Vision-Language Model performance. While our approach contributes to improved model capabilities and efficiency, we acknowledge that generating synthetic data using existing models may perpetuate or amplify societal biases present in the training data of those models. These inherited biases could affect the downstream performance and fairness of systems utilizing our methods. We encourage future work to investigate techniques for measuring and mitigating bias propagation in synthetic data generation pipelines. Additionally, practitioners implementing these methods should carefully consider the potential implications for fairness and representation in their specific applications.

## Acknowledgments and Disclosure of Funding

We sincerely thank Natasha Butt, Mazda Moayeri, Arindam Mitra, and Alessandro Stolfo for their valuable feedback and insightful discussions throughout this project. This research was partially supported by the National Science Foundation CAREER Award (Award No. 2146492), National Science Foundation Grant (Award No. 2421782), the Simons Foundation, Cisco Systems, Optum AI, the UCLA Hellman Fellowship, an Okawa Research Grant, and the Amazon Doctoral Fellowship.

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

## Appendix A. Exact Input to Stronger VLM and Generated Text Annotations

### Exact Prompt to Stronger VLM

```
You are an expert in <name of task e.g. chart understanding / diagram
    understadning / spatial reasoning >. Given example image-question-answer
    tuples,
your task is to generate diverse high-quality question-answer pairs relevant
to this skill similar to the provided examples.

Step-by-Step Process:

1. Analyze the Example: Review the provided example question-answer pair to
    understand the structure, focus, and context.

2. Understand the New Image: Infer relevant details, objects, and themes in
    the new image, considering how they relate to the skill.
3. Generate Questions: Create questions that reflect the context and
    content of the new image, ensuring they align with the skill and follow
    the example's style.
4. If the question is a multiple-choice question, make sure to include the
    options in the question.
5. Formulate Answers: Generate accurate and concise answers to the
    questions. Ensure each answer directly corresponds to the content of
    the new image.

Output Format:
Return the results as a JSON list of objects. Each object should include:
- "Q": The generated question (include options if it's multiple-choice).
- "A": The generated answer.

Example Output:
[
  {"Q": "Generated question 1", "A": "Generated answer 1"},
  {"Q": "Generated question 2", "A": "Generated answer 2"}
]

<Refererence Sample>

<Candidate Image>
```

Figure 7, Figure 8 and Figure 9 show examples generated by MM-GEN for chart understanding, diagram understanding and spatial reasoning on map, respectively.

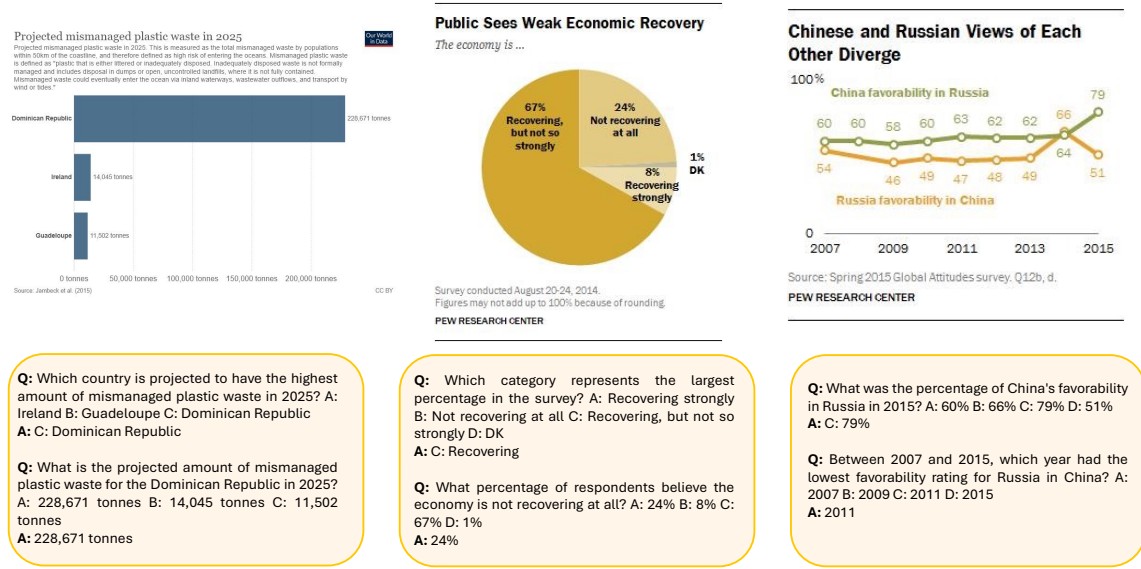

Figure 7: Examples Generated by MM-Gen for Chart Understanding

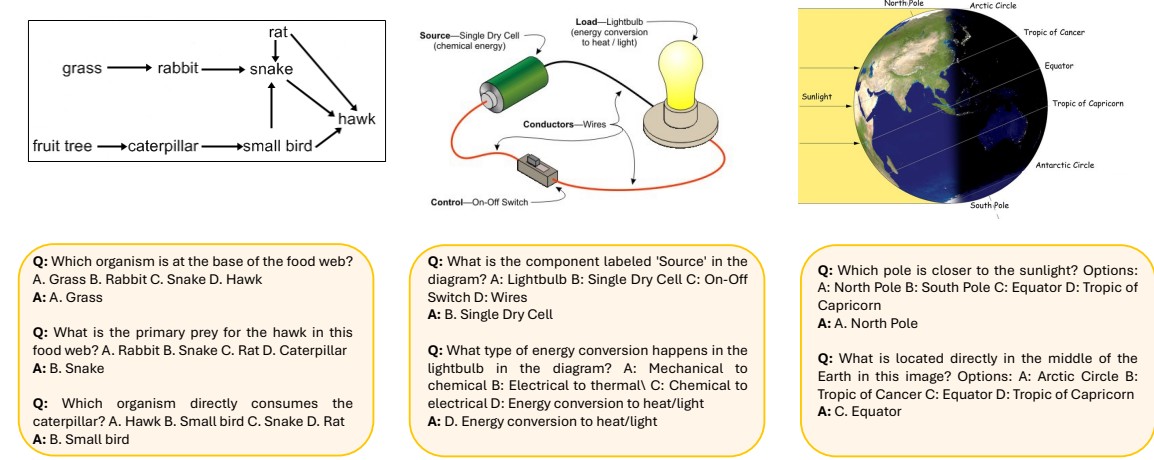

Figure 8: Examples Generated by MM-Gen for Diagram Understanding

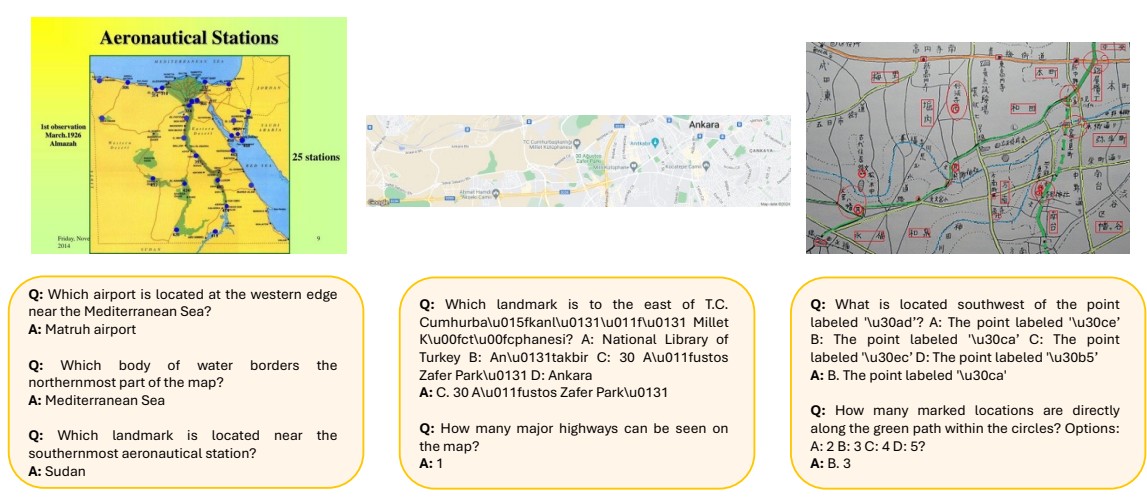

Figure 9: Examples Generated by MM-GEN for Spatial Reasoning on Maps

## Appendix B. Pseudocode for MM-Gen

In this section, we present the exact pseudocode for MM-Gen. Each of the three steps is denoted as a subroutine in the pseudocode.

---

**Algorithm 1** Data Generation Process

---

1: **Subroutine 1: Partition** (§ **??**)
2: $\{(S_{T_k}^{\text{ref}}, V_{T_k}^{\text{pool}})\}_{k \in \text{types}_T} = \text{PARTITION}(S_T^{\text{ref}}, V_T^{\text{pool}}, \text{types}_T)$
3: **Subroutine 2: Generate Data** (§ **??**)
4: **for all** $k \in \text{types}_T$ **do**
5:    $\mathcal{D}_k^{\text{GEN}} \leftarrow \emptyset$
6:    $\text{Iterator}(V_{T_k}^{\text{pool}}) \leftarrow$ Randomly order elements of $V_{T_k}^{\text{pool}}$ and create an infinite iterator
7:    Set NUM_GEN_PER_REF $\leftarrow N \cdot \frac{|S_{T_k}^{\text{ref}}|}{|S_T^{\text{ref}}|}$
8:    **for all** $(v^{\text{ref}}, t_p^{\text{ref}}, t_{\text{res}}^{\text{ref}}) \in S_{T_k}^{\text{ref}}$ **do**
9:      **for** $i = 1$ to NUM_GEN_PER_REF **do**
10:        $v_{\text{candidate}} \leftarrow \text{NEXT}(\text{Iterator}(V_{T_k}^{\text{pool}}))$
11:        $(t_p, t_{\text{res}}) \leftarrow L_{\text{VLM}}(\text{SYS\_PROMPT}, v^{\text{ref}}, t_p^{\text{ref}}, t_{\text{res}}^{\text{ref}}, v_{\text{candidate}})$
12:        $\mathcal{D}_k^{\text{GEN}} \leftarrow \mathcal{D}_k^{\text{GEN}} \cup \{(v_{\text{candidate}}, t_p, t_{\text{res}})\}$
13:      **end for**
14:    **end for**
15: **end for**
16: $\mathcal{D}^{\text{GEN}} \leftarrow \bigcup_k \mathcal{D}_k^{\text{GEN}}$
17: **Subroutine 3: Filter** (§ **??**)
18: $\mathcal{D}^{\text{GEN}_{\text{filt}}} \leftarrow$ Filter $\mathcal{D}^{\text{GEN}}$ by computing perplexity of all examples and selecting middle $r\%$ of examples
19: Return $\mathcal{D}^{\text{GEN}_{\text{filt}}}$

---

## Appendix C. Examples of Baselines Hurt Performance on Some Tasks

In Sec. 5, we observed that for ChartQA and for SpatialMap, some of the baselines can hurt the performance of the base model. Upon inspecting the generated data, we noticed that this is due to highly repetitive generated text annotations for these baselines. Here, in Figures 10, 11, 12, we present some of the generated examples to highlight this problem.

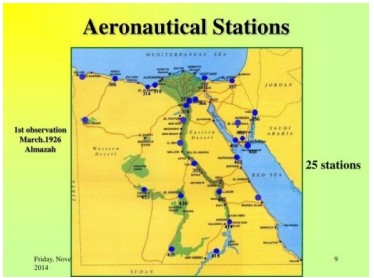

**Task-Specific Annotation Generated Using Natural Language Description**

Q: According to the map, how many aeronautical stations are shown in Egypt? A: 15 stations B: 20 stations C: 25 stations D: 30 stations
A: B. 25 stations

Q: What is the total number of aeronautical stations marked on the map? A: 20 B: 25 C: 30 D: 35
A: B. 25

Q: How many aeronautical stations are indicated on the map? A: 20 B: 25 C: 30 D: 35?
A: B. 25

Figure 10: Task: Spatial Reasoning on Maps; **Highly Repetitive** Generated Text for Baseline 3: Task-Specific Text Annotations, Generated Using Natural Language Task Description

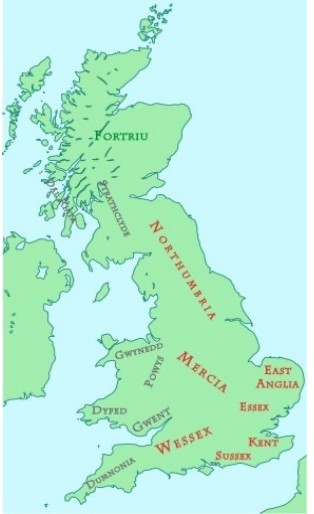

**Task-Agnostic Annotation**

The map depicts the island of Great Britain during a historical period characterized by several distinct kingdoms. In the north, the region of Fortriu is marked, suggesting a location within present-day Scotland. Moving south, Northumbria stretches across the northeastern portion, indicating its large influence during this era. To the west of Northumbria lies the domain of Rheged, near Strathclyde, hinting at complex territorial relationships. The central part of the island is dominated by Mercia, which extends extensively and is a major kingdom. In the southwest, the realms of Powys and Gwent are noted, highlighting the territorial divisions within modern-day Wales. Wessex, in the south-central region, appears prominent and influential. The southeastern territories feature East Anglia, Essex, Kent, and Sussex, each delineated as separate entities, representing the distinct Anglo-Saxon kingdoms of that time.

The map illustrates the territories of historic Great Britain during the early medieval period. Major regions are marked in red, such as Northumbria, Mercia, and Wessex, which were prominent Anglo-Saxon kingdoms. To the northeast, Northumbria extends down the eastern coast, while Mercia occupies the central part of the island stretching towards the west. Wessex is located in the south, with its influence reaching the coastal areas. Smaller regions like Kent, Sussex, and Essex are positioned along the southeastern coastline. In the western part, Welsh territories such as Gwynedd and Powys are noted, indicating a distinct cultural and political identity. Scotland is dominated by the territories of Fortriu and Pictland to the north. The map highlights the diverse and complex political landscape of the British Isles during this era.

Figure 11: Task: Spatial Reasoning on Maps; **Highly Repetitive** Generated Text for Baseline 2: Task-Agnostic Captions

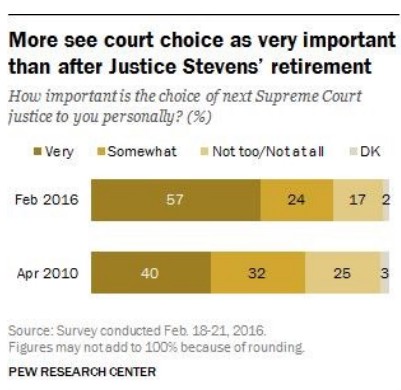

**Task-Specific Annotation Generated Using Natural Language Description**

**Q:** What percentage of respondents in February 2016 considered the choice of the next Supreme Court justice to be 'very important'? A: 57% B: 40% C: 24% D: 32%
**A:** A. 57%

**Q:** What percentage of respondents in February 2016 considered the choice of the next Supreme Court justice to be 'very important'? A: 57% B: 40% C: 24% D: 32%
**A:** A. 57%

**Q:** How did the percentage of people who found the choice of the next Supreme Court justice 'Very' important change from April 2010 to February 2016? A: Increased by 10% B: Decreased by 7% C: Increased by 17% D: Decreased by 15%
**A:** C. Increased by 17%

**Q:** How did the percentage of people who considered the choice of the next Supreme Court justice as 'Very important' change from April 2010 to February 2016? A: Increased B: Decreased C: Stayed the same D: Cannot be determined"?
**A:** A. Increased

Figure 12: Task: Chart Understanding; **Highly Repetitive** Generated Text for Baseline 3: Task-Specific Text Annotations, Generated Using Natural Language Task Description

## Appendix D. Data Generation and Training Details

For all data generation, we used the GPT-4o model OpenAI (2023) (2023-06-01-preview). For all the experiments, we use the follow common hyperparameters and trained on 4 A-100 GPUs.

Table 4: Training Hyperparameters for MM-Gen

| Hyperparameter | Value |
|---|---|
| Model Name or Path | liuhaotian/llava-v1.5-7b or liuhaotian/llava-v1.5-13b |
| Vision Tower | openai/clip-vit-large-patch14-336 |
| MM Projector Type | mlp2x_gelu |
| MM Vision Select Layer | -2 |
| MM Use Image Start/End Token | False |
| MM Use Image Patch Token | False |
| Image Aspect Ratio | Pad |
| Group by Modality Length | True |
| BF16 | True |
| Train Batch Size (Per Device) | 16 |
| Eval Batch Size (Per Device) | 4 |
| Gradient Accumulation Steps | 1 |
| Learning Rate | 2e-5 |
| Weight Decay | 0.0 |
| Warmup Ratio | 0.03 |
| LR Scheduler Type | Cosine |
| TF32 | True |
| Model Max Length | 2048 |

For each of the tasks, we tuned the number of epochs such that training loss converged for the MM-Gen generated data.

1. Chart Understanding (ChartQA): 6 epochs

2. Diagram Understanding (AI2D): 6 epochs

3. Spatial Reasoning on Map (SpatialMap): 3 epochs

