# OpenReview forum: "MM-GEN: Principled and Generalizable Data Curation for Enhancing Task Performance in VLMs"
_DMLR — Accepted by DMLR_

### Review · Reviewer_8THe · 2025-08-24

**Recommendation:** 2
**Confidence:** 1

**Summary Of Contributions:**

This paper introduces MM-Gen, an automated framework for generating task-specific text annotations to improve VLM performance on specialized visual understanding tasks. The key contribution is a systematic approach that addresses four critical data curation principles: coverage, diversity, quality, and informativeness. Through extensive experiments on chart understanding, diagram interpretation, and spatial reasoning tasks, the authors show that fine-tuning Llava-1.5 with data from MM-Gen leads to absolute performance improvements of up to 29%. The work demonstrates that a data-centric, principled approach to synthetic data generation is a highly effective method for enhancing VLM capabilities.

**Strengths:**

The framework provides a structured solution for VLM data curation by systematically addressing four key principles: it ensures comprehensive coverage across task subgroups and generates diversity in its annotations, while using perplexity-based filtering to maximise both the quality and informativeness of the data. The work is relevant as the proposed pipeline could benefit the research community.

**Audience:**

Yes

**Broader Impact Concerns:**

The paper includes a Broader Impact Statement acknowledging bias propagation risks, which is appropriate.

**Claims And Evidence:**

The paper's claims are well-supported by its experimental results. The "generalizable" claim in the title might be overstated due to limited task diversity (charts, diagrams, maps).

**Datasets And Benchmarks:**

The paper uses established benchmarks (ChartQA, AI2D, SpatialMap) appropriately. However, the generated datasets and codes should be made publicly available for reproducibility. The paper didn't include any public URLs for the access to the dataset and codes.

**Extended Submissions:**

This submission does not appear to be an extended version of a previously published work.

**Limitations:**

1. The effectiveness of MM-Gen is somehow constrained by the capabilities of the teacher VLM. Any errors, biases, or shortcomings in the teacher’s outputs are likely to propagate into the curated dataset and, in turn, into the student model. The requirement of stronger VLM could limits the framework's applicability in resource-constrained settings.

2. The framework is not fully automated, as it requires pre-defined keywords to partition data for coverage. This may remain a manual step that requires some knowledge and may be difficult for complex, unstructured tasks.

**Requested Changes:**

1. The empirical decision to retain the middle 50% of examples is shown to be effective, but the justification would be stronger with a sensitivity analysis. For instance, reporting how performance varies on one representative task (such as ChartQA) when retaining different proportions of the dataset (e.g., middle 20%, 40%, 60%, 80%) would make the claim more robust and offer practical guidance for practitioners on how to set this parameter.

2. The manual definition of types is an important step. It would be helpful to add a short paragraph discussing the process. For example, how sensitive is the model to the granularity of these types? What is the recommended strategy for a researcher approaching tasks where subgroups are not obvious?

3. While the experiments cover LLava-1.5 at 7B and 13B, they do not extend to more recent or diverse open-source VLMs such as InternVL or Qwen-VL, nor to stronger closed-source models. This leaves some uncertainty about how broadly applicable MM-Gen is across architectures and training regimes.

4. Please publicly release/provide the URLs for the dataset and codes.

**Strengths And Weaknesses:**

Strengths:
1. The paper present a clear and well-justified set of principles (coverage, diversity, quality, informativeness), providing a structured solution to the data curation problem.

2. The use of a small number of reference samples for in-context learning is an effective strategy to produce diverse, task-specific annotations with minimal human supervision. In addition, the proposed perplexity-based filtering mechanism offers an efficient way to simultaneously ensure annotation quality and maximize the informational value of the curated dataset.

3. The paper includes appropriate baselines (task-agnostic captions, natural language-specified annotations), tests across multiple model sizes (7B and 13B), and conducts meaningful ablations.

Weaknesses:
1. The method requires a manually specified list of subgroup types, which may be straightforward for structured domains such as charts but could be challenging to define for less well-categorized or open-world tasks.

2. There is no discussion of when MM-Gen fails, what types of annotations it struggles to generate, or how errors in the generated data impact downstream model performance.

3. The evaluation on more LLMs are needed, such as Qwen-VL, DeepSeek-VL-Chat.

4. The dataset, models and codes are not publicly released.

---

### Review · Reviewer_FiVE · 2025-08-28

**Recommendation:** 3
**Confidence:** 2

**Summary Of Contributions:**

The paper proposes to improve the performance a VLM's performance of target tasks by finetuning it on curated data generated by a stronger VLM. It identifies four characteristics of good samples, and designs corresponding methods for curation. Experiments show that it leads to better performance than baselines trained on uncurated data.

**Strengths:**

See Strengths

**Audience:**

Yes

**Claims And Evidence:**

Yes.

**Datasets And Benchmarks:**

Yes.

**Extended Submissions:**

N/A

**Limitations:**

See weaknesses

**Requested Changes:**

(Not critical) It seems the paper is highly related to knowledge distillation. It is better the draw the connection explicitly.

**Strengths And Weaknesses:**

Strengths:
1. The paper focuses on four important aspects of good text annotations of multi-modal samples for data curation: coverage, diversity, quality, informativeness. These observations are well-motivated and lead to reasonable designs in the curation pipeline.
2. Detailed ablation studies are conducted to validate the effectiveness of the four aspects.

Weaknesses:
1.  It needs a stronger VLM to generate samples and then curate, so the final model's capability is upper-bounded by the teacher VLM.
2. It requires to manually specify types for the images. While it might lead to better performance on specific tasks than task-agnostic methods, it also limits the versatility of the model's capability.

---

### Review · Reviewer_faCp · 2025-08-28

**Recommendation:** 3
**Confidence:** 2

**Summary Of Contributions:**

This work introduces MM-Gen, a principled framework for task-specific multimodal data curation. Contributions include: (1) showing task-aware annotations outperform task-agnostic captions; (2) demonstrating reference-sample–based generation achieves coverage and diversity; (3) proposing perplexity-based filtering for quality and efficiency; (4) building a scalable pipeline improving VLM performance up to 30% with minimal human intervention.

**Strengths:**

The submission introduces MM-Gen, a principled framework for task-specific multimodal data curation that advances prior work by explicitly addressing four key desiderata: coverage, diversity, quality, and informativeness. The contributions are significant, as the method achieves up to 29% absolute accuracy improvements on chart understanding, diagram interpretation, and spatial reasoning tasks, sometimes surpassing human-curated data. The research is thorough, including ablations, baseline comparisons, and evaluations across model sizes, which demonstrates robustness and scalability. The paper is clearly written, with intuitive explanations, formal grounding such as the perplexity formula, and thoughtful discussion of ethical implications including bias propagation in synthetic data.

**Audience:**

Yes

**Claims And Evidence:**

Yes

**Datasets And Benchmarks:**

N/A

**Extended Submissions:**

No

**Limitations:**

The evaluation is confined to three domains, as mentioned above.

**Requested Changes:**

(suggestion, not critical) The current experiments are limited to three tasks: chart understanding, diagram interpretation, and spatial reasoning. While these are strong benchmarks, the claims of generalizability would be more convincing if MM-Gen were evaluated on a wider range of domains such as document understanding, medical imaging, or real-world visual QA.

**Strengths And Weaknesses:**

The paper introduces MM-Gen, a principled framework for task-specific data curation that emphasizes coverage, diversity, quality, and informativeness. It demonstrates substantial performance gains of up to 29 percent on chart understanding, diagram interpretation, and spatial reasoning, in some cases surpassing human-curated data. The approach is scalable, requiring only small reference sets, and uses perplexity-based filtering to improve efficiency. Despite these strengths, reliance on stronger teacher models, limited task scope, potential bias propagation, and absence of human evaluation remain notable weaknesses.